# Functional Beverages in Selected Countries of Asia Pacific Region: A Review

**Lei Cong, Phil Bremer** and **Miranda Mirosa** *

Department of Food Science, University of Otago, P.O. Box 56, Dunedin 9054, New Zealand;
lei.cong@otago.ac.nz (L.C.); phil.bremer@otago.ac.nz (P.B.)
* Correspondence: miranda.mirosa@otago.ac.nz; Tel.: +64-3479-7953

**Abstract:** Functional beverages have gained increasing market share over the last decade. As the Asia Pacific region is one of the largest and most important markets for functional foods, it is critical when developing and promoting new products that food manufacturers/marketers have a good understanding of the Asia Pacific market, including the legislative requirements and consumers' perceptions of functional beverages. A literature review was undertaken to elucidate legislation criteria and consumers' perceptions of functional beverages in Asia Pacific countries. Topics reviewed included the origin and definitions of functional foods and beverages; the legislative criteria for functional foods and beverages in four representative countries—Australia, New Zealand, China, and Japan; and consumers' perceptions of functional beverages. There was no concrete definition of "functional food" or "functional beverage" region-wide and correspondingly, the legislative terms and regulatory frameworks for functional foods and beverages varied from country to country and showed divergence due to cultural differences. The systematic review of consumer perceptions of functional beverages showed that product acceptance and purchase intention for different functional beverages was heterogeneous among consumers in the Asian Pacific Region, with many factors playing a role including product attributes (e.g., functional attributes, sensory attributes, and product form) and consumer perceptions (e.g., health motivation, trust in food industry, and food neophobia). The findings from this review will help guide product development and inform marketing strategies for functional beverages targeting the Asia Pacific region by providing information on legislation and consumers' perceptions.

**Keywords:** Asia Pacific; functional foods; functional beverages; new product development

## 1. Introduction

The concept of "functional foods" is often cited as newly emerging. However, this idea is far from new, first being described in the ancient Vedic texts from India, as well as being an integral of Chinese traditional medicine. The vision to develop functional foods reflects the oriental philosophy that: "Medicine and food have a common origin" [1]. The recent emphasis on the development of foods with additional benefits emerged in Japan in the 1980s for food products fortified with constituents that possessed advantageous physiological effects [1–5]. The term "functional food" first appeared in 1993 in the journal *Nature* under the heading "Japan explores the boundary between food and medicine" [1,6].

There is no doubt that the interest of Japanese consumers in functional foods consequently increased awareness of such products worldwide. The concept of functional foods embraces the idea that food can have a role beyond gastronomic pleasure or energy and nutrient supply [7]. This renewed interest in functional foods continues to grow, powered by progressive research efforts to identify beneficial health properties and potential applications of nutraceutical substances and owing to increasing public interest in the role of food in health and wellness [8].

According to Corbo et al. 2014, functional beverages are by far the most active and popular category of functional foods, owing to their convenience and ability to meet consumer demands for desirable nutrients and bioactive compounds, ease of distribution and storage and the ability to alter the container size, shape, and appearance [9–12]. Functional beverages occupied over half (US $99 billion) [13] of the total market value (US $168 billion) of functional foods in 2019 [14], with approximately 1/3 of the market value (US $36 billion) being contributed by the Asia Pacific region [13].

To provide information to food manufacturers and marketers who are developing functional beverages for the Asia Pacific region, the current review focuses on legislation criteria and consumers' perceptions of functional beverages in countries within this region. The origin and definitions of functional foods and beverages are reviewed (Section 2); the legislative criteria of functional foods and beverages among selected countries is introduced and examples of functional beverage are presented (Section 3); and finally a systematic review of consumers' perceptions about functional beverages in the region is presented (Section 4).

## 2. General Definition of Functional Foods and Beverages

The definition of functional beverages falls under the general definition for functional foods. Functional food is essentially a marketing term and in most countries, there is no legislative definition of the term [1,2]. Although the term "functional food" has been defined several times by academic authorities, so far there is no universally accepted definition for this group of food products [15]. Very simple to more complex definitions currently in use included:

International Food Information Council (IFIC)—Foods or dietary components that may provide a health benefit beyond basic nutrition [8,16].

International Life Sciences Institute of North America (ILSI North America)—Foods that by virtue of physiologically active food components provide health benefits beyond basic nutrition [8,16].

Functional Food Science in Europe (FuFoSE), International Life Sciences Institute of Europe (ILSI Europe) & European Union (EU)—A food can be regarded as "functional" if it is satisfactorily demonstrated to positively affect one or more target functions in the body, beyond adequate nutritional effects, in a way that is relevant to either an improved state of health and well-being and/or a reduction of risk of disease. Functional foods must remain foods and they must demonstrate their effects in amounts that can normally be expected to be consumed in the diet: they are not pills or capsules, but are part of a normal food pattern [17].

Functional foods and beverages can be considered to sit at the intersection between conventional foods and beverages and pharmaceuticals (Figure 1), however drawing a line between these three is challenging, even for nutrition and food experts. Typically, a food marketed as functional contains added technologically developed ingredients with specific health benefits [18].

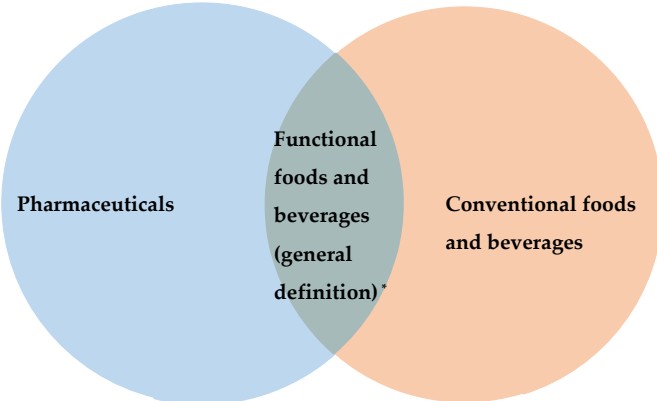

**Figure 1.** The nature of the relationship between pharmaceuticals, conventional foods and beverages, and functional foods and beverages. * Different terms describing foods and beverages with specific health benefits, such as nutraceuticals, health foods, and dietary supplements, etc., can be included in this area.

With regard to pharmaceuticals, their core definition is any article that is "intended for use in the diagnosis, cure, mitigation, treatment, or prevention of disease in man or other animals." [8]. At the same time, certain health claims can be made for foods and ingredients that are associated with health conditions. In order to distinguish pharmaceuticals and functional foods and beverages, it is important to highlight that functional foods and beverages must be foods or beverages and their beneficial effects should be obtained by consuming normal amounts of foods or beverages within the "normal" diet [7]. The benefits that consumers perceived in foods and beverages range from basic to functional with specific health benefits differentiating functional foods from conventional foods (Figure 2). In contrast to conventional foods, functional foods and beverages have demonstrated physiological benefits and can reduce the risk of chronic disease beyond basic nutritional functions [8].

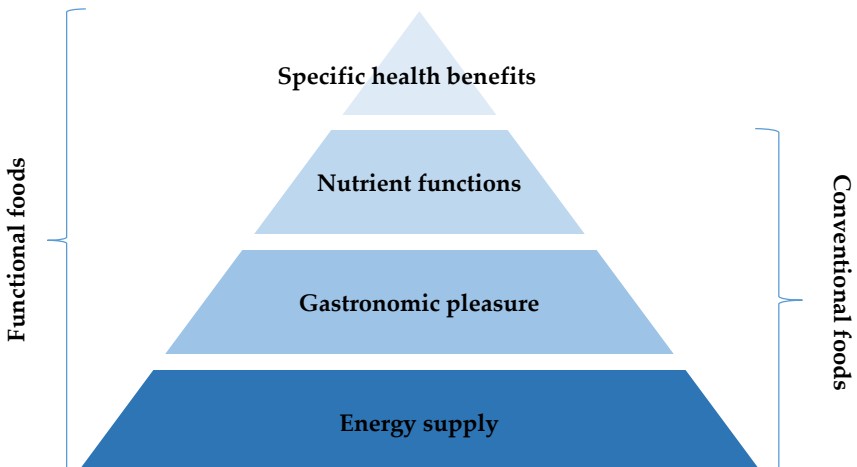

**Figure 2.** Categories of benefits from various food and beverage products as perceived by consumers.

Therefore, while there is some debate about what constitutes a functional food or beverage, there is a consensus about the term "functional" in that functional foods and beverages are used to enhance certain physiological functions in order to prevent or even cure medical conditions [19]. However, there is controversy regarding the nature of functional foods and beverages, such as (i) whether capsules, pills, and powders should be included, (ii) whether a distinct class for these products should be set up, and (iii) restrictions around allowable health claims. Despite these controversies, in a broad way, a functional food or beverage can be (i) a natural food or beverage, (ii) a food or beverage to which a component has been added, (iii) a food or beverage from which a component has been removed, (iv) a food or beverage where one or more components has been modified, (v) a food or beverage in which the bioavailability of an active agent has been modified, or (vi) any combination of the above [1].

In addition, functional beverages have previously been classified into the following three categories (i) dairy-based beverages including probiotics and minerals/$\omega$-3 enriched drinks, (ii) vegetable and fruit beverages, and (iii) sports and energy drinks [9].

## 3. Legislation Frameworks of Functional Foods and Beverages in Asia Pacific

In the absence of a concrete definition for "functional foods", a series of different terms, such as nutraceutical, health food, and dietary supplement, have contributed to the increasing confusion around this category among health professionals, food producers, legislators, and consumers. As Figure 1 illustrates, these products can be considered to lie between pharmaceuticals and conventional foods. In addition, the meaning and common usage of such terms can be significantly influenced by the legislation in different countries. Examples include "health food" in China [20], "nutraceutical" in US [7], and "Food for Specified Health Uses (FOSHU)" in Japan [21]. It is interesting to note that the legislation regarding functional foods diverges considerably between the East and the West, with cultural differences being considered to be one of the most important reasons for these differences [2].

Japan and China, for instance, consider functional foods as specific food categories, which means that after approval, particular symbols can be displayed on the food label. However, in Western countries, such as the EU, Australia, and New Zealand, functional foods are accepted as a concept, legislated by health claims. A functional foods product often means adding functionality to an existing traditional food product (often a mainstream product), and such food products do not create a new group with special symbols [2]. Typical examples of two different legislative approaches and regulations regarding functional foods in countries in the Asia Pacific region are introduced in the following section for Australia, New Zealand, China, and Japan.

### 3.1. Australia and New Zealand

A common phenomenon in western societies is that the legislation does not consider functional foods as specific food categories, but rather a concept, as in the European Union, Australia, and New Zealand. Rather than regulating the product group per se, legislative efforts in these countries are directed towards restricting the use of health claims on packs and in marketing [2,18,22]. As a case study, the current activity in health claims in Australia and New Zealand is discussed in this sub-section.

Since the passage of the Food Standards Australia New Zealand Act of 1991, these two countries have shared a common food regulatory system under the statutory agency Food Standards Australia New Zealand (FSANZ) [23]. The food standards code Standard 1.2.7—Nutrition, health, and related claims [24] regulates the current use of nutrient content claims and health claims, with the extending standard, Schedule 4—Nutrition, health, and related claims [25], providing the specifics of the claims.

Differing from nutrient content claims, health claims link a nutrient to a health effect. The standard for health claims addresses foods and the representation of their nutritional or health benefits through general or high-level claims. The approach of the standard is to remove ambiguity in the marketplace, provide a comprehensive framework, protect and assist consumers, provide opportunity for industry, have regard to costs, and work with community support [23]. The framework of FSANZ focuses on the substantiation of different types of claims in a managed system while integrating current practices and allowing for a phase-in period. Importantly, this means that not all foods are treated as equal; that is, foods must first meet eligibility criteria based on their overall nutritional profile [26]. The Nutrient Profiling Scoring Criterion (NPSC) is a nutrient profiling system used in Australia and New Zealand to determine whether a food is suitable to make a health claim, based on its nutrient profile. Only foods that meet a certain score will be allowed to have health claims made about them [27]. In short, the NPSC prevents a food that is deemed to be unhealthy, owing to its levels of say fat, sugar, or salt, to be marketed as a "health" promoting food.

The claimed framework is based on a risk assessment model that starts with a principle of "do no harm" with a defining point of not addressing serious disease. Thus, although it is acknowledged that food is essential for life, there is still much that is not known about the underlying physiological mechanisms of foods' health benefits. In contrast, disease generally requires medical treatment and health claims on food should not deter the consumer from seeking it [23].

General-level claims can be most easily understood as being health maintenance claims. For example, with regard to the nutrient Calcium, general health claims include "necessary for normal teeth and bone structure", "necessary for normal nerve and muscle function", "necessary for normal blood coagulation", "contributes to normal energy metabolism", "contributes to the normal function of digestive enzymes", "contributes to normal cell division", and "contributes to normal growth and development" [25]. According to Schedule 4, there are 200 approved general health claims for 39 nutrients.

In contrast, high-level claims require formal pre-approval. These claims are associated with more serious conditions and cover biomarker and risk reduction claims. There are currently 13 high level health claims in Schedule 4 (Table 1). In addition, a specific health effect can only be claimed with respect to requirements of a relevant population, context claim statements, and other conditions [25].

**Table 1.** Permitted high level health claims in Australia and New Zealand, extracted from Schedule 4 [25].

| Food or Property of Food | Specific Health Effect |
| --- | --- |
| A high intake of fruit and vegetables | Reduces risk of coronary heart disease |
| Beta-glucan | Reduces blood cholesterol |
| Calcium | Enhances bone mineral density<br>Reduces risk of osteoporosis<br>Reduces risk of osteoporotic fracture |
| Calcium and Vitamin D | Reduces risk of osteoporosis<br>Reduces risk of osteoporotic fracture |
| Folic acid (but not folate) | Reduces risk of foetal neural tube defects |
| Increased intake of fruit and vegetables | Reduces risk of coronary heart disease |
| Phytosterols, phytostanols, and their esters | Reduces blood cholesterol |
| Saturated fatty acids | Reduces total blood cholesterol or blood LDL cholesterol |
| Saturated and trans fatty acids | Reduces total blood cholesterol or blood LDL cholesterol |
| Sodium or salt | Reduces blood pressure |

*3.2. China*

In China, "health food" is a far better-known term than "functional food" by the public. When the first Chinese health food was approved in late 1996, the Chinese health food industry started its journey and its market value has expanded dramatically [28]. Now, China's health food market including nutrient supplements is one of the largest markets in the world [28] and was valued at US $42 billion in 2018 [29]. Moreover, consumers' mind-set about health food has shifted from seeing them as being luxury goods to ordinary consumer products [29].

According to the legislation, health foods cannot be considered as having the same scope as functional foods. Under National Standard for Food Safety—Health Food (GB 16740-2014), a health food is defined as "any food stuff claiming to have specific health functions, or to supplement nutrition with vitamins and minerals for a specific functional purpose. A health food should be designated as useful for specific consumers, can regulate bodily functions, is not designed to treat disease, and does not cause any acute, sub-acute or chronic negative effects when consumed by humans" [30]. The most recent regulations regarding this category and approval of health foods were released in Food Safety Law (FSL), October 2015, by the China Food and Drug Administration (CFDA) [20] with three material schedules, Raw Material Schedule for Health Food, Schedule of Permissible Functional Health Claims Allowed to be Used on Health Food, and Schedule of Substances Which Are Both Food and Traditional Chinese Medicines, published through FSL attracting a great deal of interest as discussed in the following sections.

Raw Material Schedule for Health Food. The CFDA's schedule of raw material includes the names of the raw materials, daily usage, and health functions. Article 75 of FSL states that "raw materials included in the schedule shall be used for the production of health food only and may not be used for the production of other food" [20]. A further explanation specifies that raw materials included in the schedule, with regulated usage and claiming corresponding health functions, can only be used for the production of health food [31]. Additionally, based on this schedule, the requirements of product approval methods are distinguished. Filing with CFDA is required for health foods using raw materials within the schedule and nutrient supplements (vitamins and minerals, etc.) imported into China for the first time. Meanwhile, health foods that use raw materials not listed in the schedule and health food products imported for the first time (excluding nutrient supplements of vitamins and minerals, etc.) are required to be registered in CFDA [20]. Specifically, the regulatory requirements about health foods fall into four categories as listed in Table 2.

**Table 2.** Regulatory requirements of health food products in different categories after October 2015 in China [20].

| Category | Approval Method | Governmental Section |
|---|---|---|
| Health foods using raw materials within the schedule | Filing | Provincial FDA |
| Nutrient supplements of vitamins and minerals, etc. imported into China for the first time | Filing | CFDA |
| Health foods that use raw materials not listed in the schedule | Registration | CFDA |
| Health food products imported for the first time (excluding nutrient supplements of vitamins and minerals, etc.) | Registration | CFDA |

To date, only part one of the schedule, regarding raw materials of nutrient supplements, has been published and implemented [32].

Schedule of Permissible Functional Health Claims Allowed to be Used on Health Food. According to FSL, health foods can claim health functions, which should be supported by scientific evidence and cannot cause any acute, sub-acute, or chronic negative effects on human bodies. CFDA, collaborating with other relevant governmental sections, is responsible for planning, adjusting, and publishing the Schedule of permissible functional health claims allowed to be used for health foods [20]. In other words, all health claims for health foods are required to comply with this schedule. Exported food products that use CFDA's schedule of raw materials for health food, as well as CFDA's schedule of permissible functional health claims, will be considered health foods and are required to have scientific evidence to support the functional health claims that are made [33]. Based on Seeking Comments Regarding Administration of Health Functions Claimed by Health Food (draft for comments), released for periodic reviews by CFDA in December 2016, health claims include two categories: nutrient supplements claims, described as "supply XXX", and general function claims. The latter indicates claims of assisting health maintenance and improvement, without involving diseases [34]. To date, only the first category, functional health claims of nutrient supplements, have been published and implemented [32]. Part two of the schedule is yet to be published. It should be noted that previous claims (Table 3), published by the Chinese Ministry of Health, will likely be superseded by the new schedule about functional health claims.

**Table 3.** Health claims for functional foods published by Chinese Ministry of Health [33,35].

| | Health Claims for Functional Foods |
|---|---|
| 1 | Immune regulation |
| 2 | Regulate blood lipids |
| 3 | Regulate blood sugar |
| 4 | Delay aging |
| 5 | Improve memory |
| 6 | Improve vision |
| 7 | Promote lead discharge |
| 8 | Clear and moisten throat |
| 9 | Regulate blood pressure |
| 10 | Improve sleep |
| 11 | Promote lactation |
| 12 | Anti-mutation |
| 13 | Anti-fatigue |
| 14 | Anti-hypoxia |
| 15 | Anti-radiation |
| 16 | Lose weight |
| 17 | Promote growth and development |
| 18 | Improve osteoporosis |
| 19 | Improve nutritional anemia |
| 20 | Aiding protection against chemical liver injury |
| 21 | Beauty (acne/removing melasma/improve skin moisture and oil) |
| 22 | Improve gastrointestinal function (regulate intestinal flora/promote digestion/laxative/auxiliary protection of the gastric mucosa) |

Schedule of Substances Which Are Both Food and Traditional Chinese Medicines. The Chinese population has a deeply entrenched appreciation for the health benefits that can be obtained from foods [36]. According to Chinese culture, diet and lifestyle impact on current health and are important predicators of future health. Underpinning this overall cultural appreciation of the prophylactic and therapeutic properties of foods is the extraordinarily complex system of traditional Chinese medicine (TCM) [37]. Over the history of TCM use, some substances that have also been used for foods for a considerably long time can be used as herbs in TCM prescriptions. When producing foods, these substances should not be considered as medicine since they are foods originally. Based on this tradition, a schedule regulating substances that are both foods and traditional Chinese medicines will be released by the National Health and Family Planning Commission of China. Currently, a draft version is available for comment and there are 101 substances in total in the schedule [38]. According to FSL, medicines cannot be included in food products, whereas substances considered both food and TCM herbs are excluded from this regulation [20].

In conclusion, the Raw Material Schedule for Health Food is the most important criterion for categorizing health foods. Meanwhile, all health claims on foods are required to comply with the Schedule of Permissible Functional Health Claims Allowed to be Used on Health Food. Although medicines cannot be included in food products, substances that are considered to be both food and TCM herbs may be involved. Following FSL, it is easy to find that legislative regulations are focusing on health foods. However, based on health claims, health foods cannot cover all of the functional foods. Foods with substances considered to be both food and TCM herbs, and even other foods, may be parts of functional foods. Excluding health foods, National Food Safety Standard for Nutrition Labelling of Pre-packaged Foods [39] regulates 65 standard nutrient function claims for prepacked food products. Categories with regard to functional foods and the legislative criteria of their health claims are summarized in Table 4.

**Table 4.** Categories with regard to functional foods and the legislative criteria of their health claims.

| Category | Legislative Criteria of Functional Materials | Legislative Criteria of Health Claims |
|---|---|---|
| Health foods | Raw Material Schedule for Health Food | Schedule of Permissible Functional Health Claims Allowed to be Used on Health Food |
| Foods with substances considered both food and TCM herbs | Schedule of Substances That Are Both Food and Traditional Chinese Medicines | National Food Safety Standard for Nutrition Labelling of Prepackaged Foods |
| Other foods (excluding special foods) | N/A | National Food Safety Standard for Nutrition Labelling of Prepackaged Foods |

Approved functional foods are authorized by the CFDA to bear the health food certification commonly known as the Blue Hat logo (Figure 3). The packs of the beverage Red Bull, a common energy drink worldwide, in Japan, China, Australia, and New Zealand illustrate differences in label information. As shown in Figure 4, a clear symbol for health foods (Blue Hat) is seen on the Chinese pack, whereas a similar symbol is not used on the packs from other countries.

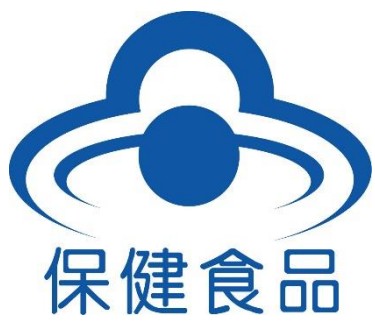

**Figure 3.** Symbol authorised by CFDA for products of health foods.

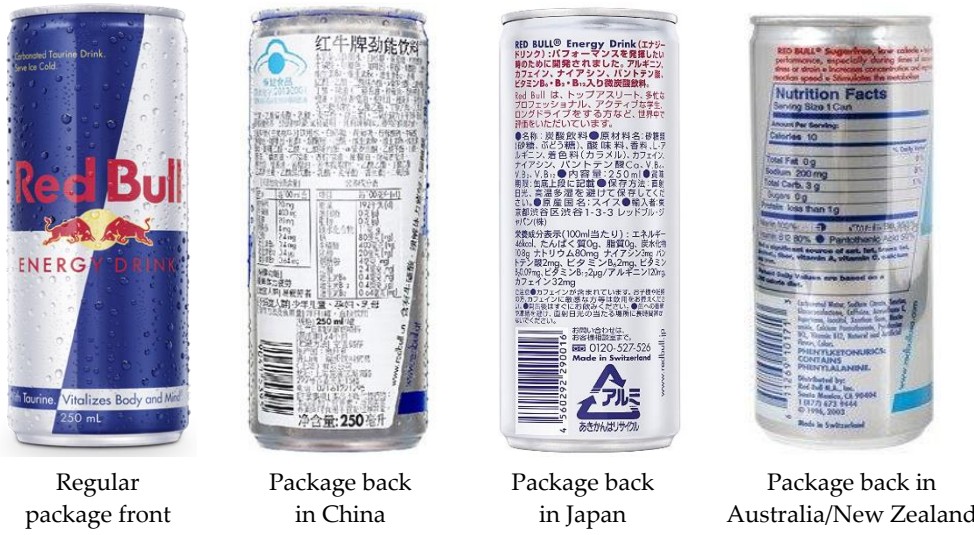

| Regular package front | Package back in China | Package back in Japan | Package back in Australia/New Zealand |

**Figure 4.** A commercial package example of the beverage Red Bull in Japan, China, and Australia/New Zealand.

CFDA has opened its dynamic database so that the general public can search for information about specific domestic or imported health food products including claims made, ingredients, and country of origin [40]. As of January 2020, CFDA had approved a total of 17,947 health food products, of which 17,164 were domestically made and 783 were imported. It is important to note that "enhancing immunity" was the most frequently approved health claim [40]. Similarly with Japan, health foods in China are considered as one of the three kinds of special foods, together with formulas for special medical uses and infant formulas [20].

### 3.3. Japan

Japan was the first country to legislate functional food products [7]. In 1991, the Japanese Ministry of Health introduced rules for the approval of a specific health-related food category called Food for Specified Health Uses (FOSHU), which included the establishment of specific health claims for this type of food [41]. Today, a more detailed category is applied to Food with Health Claims (FHC), regulated by the Ministry of Health, Labour, and Welfare (MHLW) [42]. Table 5 illustrates two categories of FHC, (i) Foods with Nutrient Function Claims (FNFC): foods that are labelled with functions of nutritional ingredients (vitamins and minerals) and (ii) FOSHU: foods officially approved to claim their physiological effects on the human body.

**Table 5.** Categories of Medicine, Food with Health Claims, and Other Food regulated by MHLW [42].

| Medicine | Food with Health Claims (FHC) | | Other Food |
|---|---|---|---|
| | Foods with Nutrient Function Claims (FNFC) (standard regulation system) | Food for Specified Health Uses (FOSHU) (individual approval system) | (some may be included in functional foods) |

A general consensus in academia is that FOSHU is the initial concept of functional foods [1,2,7]. However, Table 5 obviously presents a wider and more comprehensive concept about functional foods in Japan nowadays, that is the scope of functional foods is larger than FOSHU and even larger than FHC. Meanwhile, FOSHU could be considered as a typical kind of functional food.

Japanese law regulates functional foods by both establishing specific legislative food categories and placing a restriction on health claims that can be made on packages. Any foods that comply with specifications and standards by MHLW and are labelled with certain nutritional or health functions are categorised as FNFC. Meanwhile, only products for which approved health claims can be made (Table 6) and that meet certain requirements [21] can have the FOSHU symbol on their packages (Figure 5). Packs of Yakult [42], a product that can modify gastrointestinal conditions and is marketed in Japan, China, and Australia/New Zealand, show examples of different label information (Figure 6). Note the occurrence of a clear symbol for FOSHU on the Japanese pack, whereas a similar symbol is not present on the packs for other countries.

**Table 6.** Claims permitted under FOSHU criteria [21].

| **Approved FOSHU Products** |
|---|
| Foods to modify gastrointestinal conditions |
| Foods related to blood cholesterol level |
| Foods related to blood sugar levels |
| Foods related to blood pressure |
| Foods related to dental hygiene |
| Cholesterol plus gastrointestinal conditions, triacylglycerol plus cholesterol |
| Foods related to mineral absorption |
| Foods related to osteogenesis |
| Foods related to triacylglycerol |
| **Approved reduction of disease risk claim** |
| Calcium and Osteoporosis |
| Folic Acid and Neural Tube Defect |

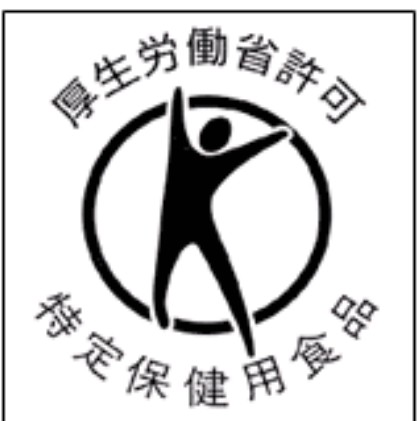

**Figure 5.** Symbol for FOSHU approval in Japan.

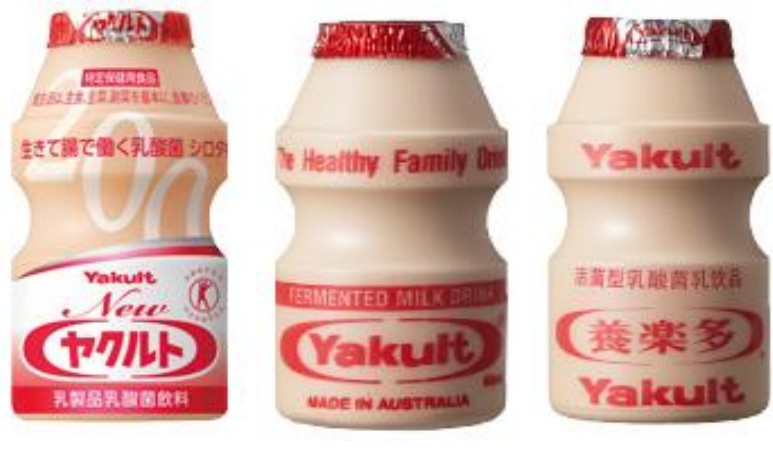

Japan　　　　　　　Australia/New Zealand　　　　　　　China

**Figure 6.** Commercial packs of the beverage Yakult in Japan, China, and Australia/New Zealand.

FOSHU is considered as one of the five kinds of Food for Special Dietary Uses (FOSDU), together with: formulas for pregnant or lactating women; infant formula; food for the elderly with difficulty in masticating or swallowing; and mdicinal foods for the ill [43]. In 2001, FOSHU products in Japan were allowed to take the form of capsules and tablets, although the great majority of products are still in conventional food formats [44].

## 4. Consumers' Perceptions of Functional Beverages in Asia Pacific

*4.1. Method*

### 4.1.1. Research Questions

Consumers' perceptions have had a large impact on the development of the functional beverage market. Considering the differences in geography and culture, it is crucial to understand consumers in the target market. Therefore, in order to provide consumer-oriented information to food manufacturers and marketers who are working with the development of functional beverages for the Asia Pacific region, a systematic review on Asia Pacific consumers' perceptions upon functional beverages was conducted. Two research questions were formulated at the beginning: (i) What is the acceptance and purchase intention for functional beverages? (ii) What factors influence consumers' perceptions of functional beverages?

### 4.1.2. Data Collection

A literature search of the Web of Science (Core Collection) was conducted in November 2019. By utilizing the method developed by Hartmann and Siegrist [45], a five step data collection process was applied:

1.  Search articles by using the string—("perception" or "acceptance") and (beverage * and function * and consum *) on titles, abstracts, and keywords. This search resulted in 180 records.
2.  Screen articles one by one on their titles, abstracts, and keywords followed five inclusive criteria—(i) quantitative or qualitative studies; (ii) full-text articles published in peer-reviewed journals in the English language; (iii) not review articles, opinion paper and outlooks, conference papers and abstracts, concept articles; (iv) articles on consumer's perceptions and acceptance about non-alcoholic functional beverages, including dairy-based beverages, vegetable and fruit beverages, and sports and energy drinks; (v) articles written in English. This screen resulted in 16 articles.

3. Further select articles by the five inclusive criteria throughout full texts. This selection resulted in 12 articles.
4. Supply articles by checking reference lists of the 12 selected articles. 14 articles were identified on top of the existing 12 articles. Therefore, 26 articles were collected at this stage.
5. Determine the collected articles by reviewing whether the research focused on the Asian Pacific Region. Based on this criteria 20 articles were excluded because the research was not conducted in countries within the Asian Pacific region. In total, 6 articles were finally selected.

With regard to the research questions, information was extracted from the six relevant articles on (i) study author and publication year; (ii) methodology utilised in each study; (iii) characteristics of the study participants, including country of residence and size etc.; (iv) main findings of each study was summarised (Table 7).

*4.2. Results and Discussions*

The six studies fitting the selection criteria were carried out in six different countries/regions in the Asia Pacific region, namely Australia, Taiwan, Korea, China, Malaysia, and New Zealand (Table 7). It was interesting to note that all countries/regions covered had a relatively high Gross Domestic Product (GDP) for the Asia Pacific region [46], which suggests a link between income and a developing functional beverage market. All of the articles had been published within the last ten years, from 2009–2019, which is in line with this being an emerging area of interest. Beverages studied included dairy-based beverages (e.g., functional yogurt and milk), vegetable and fruit based beverages (e.g., functional juice, coffee, berry beverages), and functional water.

**Table 7.** Study summary and main findings from collected literatures.

| | Study | | | | Main Finding | |
|---|---|---|---|---|---|---|
| No | Author, Year | Country | Methodology | Beverage | Consumers' Acceptance/Purchase Intention | Factor of Influence |
| 1 | Malek, Umberger, Zhou, Makrides, & Huynh, 2019 [47] | Australia | Discrete choice experiment (*n* = 857) | Fortified water, juice, milk, and yogurt | There was no demand for pregnancy supplements in the form of nutritionally fortified food or beverages in the Australian pregnancy supplement market. | Product form was the strongest driver for pregnant women choosing dietary supplements. Consumers' preferences for product form were heterogeneous. |
| 2 | Wang & Yu, 2016 [48] | Taiwan | Survey (*n* = 401) | Ready-to-drink coffee beverages | Utilitarian value was one of the most crucial determinants of repurchase intentions. | Functional attribute beliefs had a dominant influence on utilitarian value. Sensory attribute beliefs had the most influence on hedonic value. |
| 3 | Kim & Kwak, 2015 [49] | Korea | Consumer acceptance test (*n* = 80) | Blueberry functional beverages | Functional information positively affected consumers' acceptability and purchase intention of functional food products, particularly when the sensory quality was within an acceptable range. | Health-related claims and a thick mouthfeel positively influenced the product's acceptability and purchase intention of blueberry functional beverages. |
| 4 | Siegrist, Shi, Giusto, & Hartmann, 2015 [50] | China & Germany | Online survey (*n*China = 443, *n*Germany = 502) | Functional yogurt and nonalcoholic beverages | Consumers in China were more willing to purchase functional foods, compared to their German counterparts. | Participants with a higher health motivation and more trust in the food industry reported a higher willingness to buy functional foods than participants with lower health motivation and less trust in the industry. Food neophobia had a negative impact on the acceptance of functional foods in the Chinese sample. |
| 5 | Chelliah, Kwon, Annamalah, & Munusamy, 2013 [51] | Malaysia | Survey questionnaire (*n* = 100) | Herbal coffee (Tongkat Ali) | Customer retention was not dependent on price and place, but was dependent on product and promotion in the marketing mix. | Customer preference, positive customer experience, satisfaction, and lasting customer loyalty were factors that impacted on the relationship between marketing mix and customer retention. |
| 6 | Jaeger, Axten, Wohlers, & Sun-Waterhouse, 2009 [52] | New Zealand | Descriptive sensory analysis (*n* = 12); Consumer acceptance test (*n* = 392) | Polyphenol-rich beverages (extracts from berry fruits and/or cocoa) | The cocoa and berry fruit formulations were less liked by consumers than formulations of berry fruit polyphenol with added sucrose. | Bitterness and chalkiness were two key sensory attributes to impact on consumers' acceptance and required optimisation. Disclosure of a beverage's health benefit to consumers may enhance its appeal to consumers. |

4.2.1. Acceptance and Purchase Intention of Functional Beverages

Consumers' acceptance and purchase intentions for different functional beverages were heterogeneous. For example, pregnant women in Australia were not willing to adopt new functional food or beverage products to fortify nutrition during pregnancy [47]. However, other studies reported that functional beverages were generally accepted by consumers, such as the study comparing China and Germany, which, in particular, highlighted that Chinese consumers had a relatively high willingness to buy functional foods [50].

Culture/geography also played a significant role in the acceptance of functional beverages. The research conducted in Malaysia focused on a herbal coffee, Tongkat Ali, which had gained high acceptance among Malay males so that customer retention was not dependent on "Price" and "Place", but on "Product" and "Promotion" in the marketing mix [51]. However, it was hard to imagine that this would be true for a market whose consumers were completely unfamiliar with this herbal coffee.

Given the low number of publications selected in the current study, it is difficult to draw many general conclusions regarding consumers' acceptance and purchase intentions for functional beverages, except to infer that perceptions differ based on differences in consumer's attitudes and product attributes. This observation is in line with previous research with Western consumers which reported that German consumers' preferences depended on their attitude to functional foods, with functional food skeptics preferring non-functional dairy beverages, while functional food advocates preferred functional dairy beverages [53]. Further, it has been reported that while most UK consumers preferred the sensory characteristics of conventional juices, a small segment of the population significantly preferred the sensory attributes of functional juices [54]. In general, the reported studies on either Asia-Pacific or European consumers highlight the complexity of consumers' acceptance and purchase intentions towards functional beverages and illustrate the fact that these cannot be summarized by a simple trend.

4.2.2. Possible Factors That Influence Consumers' Perceptions of Functional Beverages

Many factors were found to influence consumers' acceptance and purchase intention of functional beverages, including functional attributes, sensory attributes, consumer perceptions (e.g., health motivation, trust in food industry, and food neophobia), and product format.

It is obvious that a functional attribute is the main feature that distinguishes functional beverages from conventional beverages and, therefore, most studies have focused on consumers' perceptions of functional attributes, such as the impact of health-related claims on consumer acceptability [49] and purchase intentions, or the effect of exposing health information on consumers' perceived value (e.g., utilitarian and hedonic value) [48]. In Table 7, studies 2, 3, and 6 all reported that effectively disclosing a products' health benefits to consumers positively influenced consumers' acceptance and purchase intention [48,49,52]. These findings were in-line with studies in Western countries, whereby the functional attributes or health benefits were important for functional dairy beverages in Germany [53] and probiotic juice beverages in the UK [55].

In Taiwan, sensory attributes have been reported to be another key factor influencing consumers' acceptance of a coffee functional beverages [48]. Kim & Kwak, 2015 report that Korean consumers preferred a thick mouthfeel for blueberry functional beverages [49], and Jaeger et al. 2009 believed bitterness and chalkiness were two important factors that reduced New Zealand consumers' acceptance of berry fruits and/or cocoa beverages containing enhanced levels of polyphenols [52]. Similarly, in Spain, the bitterness of a functional juice beverage has been reported to adversely impact on product acceptance [56].

Health motivation and trust in the food industry positively influence Chinese consumers' willingness to buy functional foods/beverages, whereas food neophobia played a negative role in the acceptance of functional foods [50]. Other factors impacting on consumer perception of functional beverages include product form being important for Australian consumers [47].

## 5. Conclusions

Although no concrete definition of "functional food" or "functional beverage" is agreed upon worldwide, there is a consensus about the word "functional" in that functional foods and beverages are used to enhance certain physiological functions in order to prevent or even to cure diseases. The regulatory frameworks of functional foods and beverages vary from country to country, with considerable divergence occurring, owing to culture differences. For example, the Eastern societies (e.g., China and Japan) consider functional foods and beverages as specific food categories, while in many Western countries, functional foods and beverages are accepted as a concept, legislated by health claims.

Acceptance and purchase intention of different functional beverages were heterogeneous among consumers in the Asian Pacific Region, with consumer demand and culture playing significant roles. Factors that influenced consumers' acceptance and purchase intention included functional attributes, sensory attributes, consumer perceptions (e.g., health motivation, trust in food industry, and food neophobia), and product format. This observation highlights that innovators should develop products based on a deep understanding of the target consumers.

This manuscript provides information on the legislative criteria and consumers' perception upon functional beverages in the Asia Pacific Region and will contribute to industry based product development and to the currently limited academic literature on this topic.

## 6. Limitations of the Study

The research presented in this systematic review was sourced solely from the Web of Science database and it is likely that additional studies would have been found if other databases such as Scopus were also screened.

## 7. Future Directions of the Research

A challenge with functional beverages is that owing to differences in legislation between countries, a single product may be categorized and consequently regulated differently in two countries. Therefore, professional interpretation of regulations is critical as is effective communication between regulators and product manufacturers in order to ensure that regulations are adhered to. For example, the detailed interpretation of European policies on the safety of food products and the protection of consumer interests by Bondoc. 2016 has played a positive role in the enforcement of food safety legislation within the European Union, especially in the Romanian region [57–60]. Similar research within the Asia Pacific Region is recommended.

In addition, consumer-oriented studies regarding functional beverages in the Asia Pacific Region are still in the early stages. Many topics can be explored in further research, such as "What health claims for functional beverages are important to consumers?" and "What is the best way to communicate the benefits of functional foods to consumers?"

**Author Contributions:** Writing—original draft preparation, L.C.; writing—review and editing, P.B. and M.M. All authors have read and agreed with the published version of the manuscript.

**Funding:** This research was funded by a University of Otago Doctoral Scholarship.

**Conflicts of Interest:** The authors declare no conflict of interest.

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
