# Peer review of "Functional Beverages in Selected Countries of Asia Pacific Region: A Review"

_beverages, doi:10.3390/beverages6020021_

Round 1

Reviewer 1 Report

When reviewing scientific papers for publication, I usually start with a general overview in terms of a structure, abstract, literature review, methodology, findings of the research, discussion, conclusions, as well as limitations of the study.

The reviewed paper entitled “Functional Beverages in Selected Counties of Asia Pacific Region: A Review” is generally structured in a proper way. There are, however no sections ‘limitations of the study’, and ’future directions of the research”. These sections should be added too, given this is a research paper.

The literature review is quite good and is strongly founded in the existing literature of the topic. Generally I claim that Author (s) provide solid theoretical foundations for the analysis using appropriate references. I would, however, recommend to add some references devoted to the latest literature associated with the topic in question (including SCOPUS papers).

One should emphasize that the whole paper is very coherent and particular sub-parts fit together.

Additionally, one can see a smooth movement from one point to the another (end of deliberations in one sub-chapter creates also a beginning of a discussion in the next one).

The weakest point of this article is the "Results and discussion" section. Discussion is interpretation of results and presents implications of the results and should follow the results. Provide the response to the research question(s). Interpret results taking into account alternative explanations - where applicable. What are the practical implications (and theoretical –where applicable) suggested by the results of your research. Include all limitations: this does not weaken your study, but adds to your credibility. Future directions for research (incompletely answered questions) often derived from limitations. New questions which emerge from your research. Be careful not to “go beyond” your data and results, in particular if the focus of your study is narrow. You can “suggest”, or even “speculate” in the discussion, but it must be clearly evident what is derived from a result and what is your suggestion, comment or speculation ... You may include a comparison with results of other similar/ compatible studies .

Generally my opinion is positive. Though I have some minor remarks which may improve the paper.

Author Response

We would like to take this opportunity to thank the Editor and Reviewers for their useful suggestions on how to improve our manuscript. We are confident that this version is much improved as a result of their feedback and comments and hope you agree so too. 

N.B. The line numbers are based on the revised tracked version (with “all makeup”).

Comment 1:

The reviewed paper entitled “Functional Beverages in Selected Counties of Asia Pacific Region: A Review” is generally structured in a proper way. There are, however no sections ‘limitations of the study’, and ’future directions of the research”. These sections should be added too, given this is a research paper.

Response:

Two new sections - ‘6. Limitations of the study’ and ‘7. Future directions of the research’ have been added at the end of the article, which can be found in Lines 424-440 and below:

Limitations of the study

The research presented in this systematic review was sourced solely from the Web of Science, database and it is likely that additional studies would have been found if other databases such as Scopus were also screened.

Future directions of the research

A challenge with functional beverages is that owing to differences in legislation between   countries, a single product may be categorized and consequently regulated differently in two countries. Therefore, professional interpretation of regulations is critical as is effective communication between regulators and product manufacturers in order to ensure that regulations are adhered to. For example, the detailed interpretation of European policies on the safety of food products and the protection of consumer interests by Bondoc I. 2016 has played a positive role in the enforcement of food safety legislation within the European Union, especially in the Romanian region [57-60]. Similar research within the Asia Pacific Region is recommended.

In addition, consumer-oriented studies regarding functional beverages in the Asia Pacific Region are still in the early stage. Many topics can be explored in further research, such as “What are desirable health claims of functional beverages perceived by consumers?” and “How to communicate the functional benefits to consumers effectively and efficiently”?

Comment 2:

The literature review is quite good and is strongly founded in the existing literature of the topic. Generally, I claim that Author (s) provide solid theoretical foundations for the analysis using appropriate references. I would, however, recommend to add some references devoted to the latest literature associated with the topic in question (including SCOPUS papers).

The weakest point of this article is the "Results and discussion" section. Discussion is interpretation of results and presents implications of the results and should follow the results. Provide the response to the research question(s). Interpret results taking into account alternative explanations - where applicable. What are the practical implications (and theoretical –where applicable) suggested by the results of your research. Include all limitations: this does not weaken your study, but adds to your credibility. Future directions for research (incompletely answered questions) often derived from limitations. New questions which emerge from your research. Be careful not to “go beyond” your data and results, in particular if the focus of your study is narrow. You can “suggest”, or even “speculate” in the discussion, but it must be clearly evident what is derived from a result and what is your suggestion, comment or speculation ... You may include a comparison with results of other similar/ compatible studies.

Response:

New discussion has been added to further respond to each of the research questions. Results have been compared with the results of other similar/compatible studies that have published recently (including SCOPUS papers), thereby addressing the point about adding new references. New text can be found in Lines 362-373 & Lines 374-403 and below:

4.2.1 Acceptance and purchase intention of functional beverages

Given the low number of publications selected in the current study, it is difficult to draw many general conclusions regarding consumers’ acceptance and purchase intentions for functional beverages, except to infer that perceptions differ based on differences in consumer’s attitudes and product attributes. This observation is in line with previous research with Western consumers which reported that German consumers’ preferences depended on their attitude to functional foods, with functional food skeptics preferring non-functional dairy beverages, while functional food advocates preferred functional dairy beverages [53]. Further, it has been reported that while most UK consumers preferred the sensory characteristics of conventional juices, a small segment of the population significantly preferred the sensory attributes of functional juices [54]. In general, the reported studies on either Asia-Pacific or European consumers, highlight the complexity of consumers’ acceptance and purchase intention towards functional beverages, and illustrate the fact that these cannot be summarized by a simple trend.  

4.2.2 Possible factors that influence consumers’ perceptions of functional beverages

Many factors were found to influence consumers’ acceptance and purchase intention of functional beverages, including functional attributes, sensory attributes, consumer perceptions (e.g. health motivation, trust in food industry and food neophobia) and product format.

It is obvious that a functional attribute is the main feature that distinguishes functional beverages from conventional beverages, and therefore, most studies have focused on consumers’ perceptions of functional attributes, such as the impact of health-related claims on consumer acceptability [49] and purchase intentions, or the effect of exposing health information on consumers’ perceived value (e.g. utilitarian and hedonic value) [48]. In Table 7 studies 2, 3 and 6 all reported that effectively disclosing a products’ health benefits to consumers positively influenced consumers’ acceptance and purchase intention [48-49, 52]. These findings were in-line with studies in Western countries, whereby the functional attributes or health benefits were important for functional dairy beverages in Germany [53] and probiotic juice beverages in UK [55].

In Taiwan, sensory attributes have been reported to be another key factor influencing consumers’ acceptance of a coffee functional beverages [48]. Kim & Kwak, 2015 report that Korea consumers preferred a thick mouthfeel for blueberry functional beverages [49], and Jaeger et al. 2009 believed bitterness and chalkiness were two important factors which reduced New Zealand consumers’ acceptance of berry fruits and/or cocoa beverages containing enhancing levels of polyphenols [52]. Similarly, in Spain the bitterness of a functional juice beverage has been reported to adversely impacted on product acceptance [56].

Health motivation and trust in the food industry positively influence Chinese consumers’ willingness to buy functional foods/beverages, whereas, food neophobia played a negative role in the acceptance of functional foods [50]. Other factors impacting on consumer perception of functional beverages include product form in Australia [47].

Reviewer 2 Report

Esteemed Authors,

It has been a great honor, as well as a pleasantly challenging activity, to review the article entitled Functional Beverages in Selected Counties of Asia Pacific Region: A Review”.

The concept of functional foods was born in Japan in the 1980s. They are foods that were developed specifically to promote health or reduce the risk of disease. Functional foods have not already been defined by the legislation in Europe. Generally, they are considered as those foods which are intended to be consumed as part of the regular diet and which contain biologically active components which offer the potential of enhanced health or reduced risk of disease. Attention concerning this category of foods has grown, new products have appeared in the European market, and interest has turned to define the standards and guidelines for the development and promotion of this kind of food.

In the European Union (EU), there is harmonized legislation on health claims, while compounds, ingredients, plants are still regulated only at the national level.

Worldwide, consumers are becoming more interested in the relation between food and health. To harmonize regulation on foodstuffs throughout the EU, the Regulation EC1924/2006 on nutrition and health claims came into force, as a first specific set of EU legal rules dealing with nutrition and health claims. A Union List of EU-wide approved applications is now being developed that creates a level playing field on which food operators can innovate, backed by legal certainty, to ultimately bring benefits to the consumer. Food innovation is perceived as a collective effort of a variety of actors within the context of a network of institutions whose activities and interactions initiate, import, and diffuse innovations. It seems that the new regulatory regime maybe not only restrictive but also selective for future functional food innovative activities.

From this point of view, the paper is of high value due to its original character, and it treats a specific subject that is of high interest for the domain of food, nutrition, innovation, food safety, and public health. With some minor exceptions (which refers to some descriptions necessary), all materials and methods are specified and described adequately.

All iconographic materials – seven tables and six figures - were given accurate descriptions, the results were described in great detail, and the conclusions are adequate. Even though the study does display certain limitations, the approach to the topic itself is a solid one, well-argued and unequivocal.

The paper is well structured and possesses a high novelty character. The major components of the article - Introduction; General definition of functional foods and beverages; Legislation frameworks of functional foods and beverages in Asia Pacific; Consumers’ perceptions of functional beverages in Asia Pacific, and Conclusion - are organized judiciously and in direct connection one with another.

The documentation is adequate, and all the authors are cited in the text of the paper. The authors of the article need to pay more attention to writing (editing of text): the existence of some small writing errors (errors of editing) makes it harder to check the citations (checking the authors from the bibliographic reference list), and it can create some confusion in terms of understanding specialized terms.

The provided scientific results are exact and precise. The goal of the conducted research is well specified and delineated. The work protocol is appropriate, and the used analysis methods are coherent with the proposed objectives.

Nevertheless, the detailed analysis of the paper has also highlighted some aspects that require revision, as follows below:

The bibliography is relevant, but presents some minor lacks when it comes to citations or mentions. To clarify some aspects, I would suggest that the authors write the bibliography evenly: for example, journal papers require either the complete journal name, or the JCR abbreviation (in the case of ISI indexed or rated journals), or the ISO abbreviation (for BDI indexed journals); moreover, for journals, I suggest that the volume, number, and pages (as the case requires) be mentioned.

For example – page 15, lines 415-416, number 6 in the bibliographic references list: Swinbanks D., O'Brien J., 1993. Japan explores the boundary between food and medicine. Nature (or JCR Abbreviation – Nature), 364, 6434, Article number: 180; DOI: https://doi.org/10.1038/364180a0.

Another example – page 16, lines 477-478, number 37 in the bibliographic references list: O'Brien P., 2015. Regulation of functional foods in China: A framework in flux. Regulatory Rapporteur (or ISO Abbreviation – Regul. Rapp.), 12, 7/8, 1-5.

Another example – page 17, lines 507-508, number 51 in the bibliographic references list: Chelliah S., Kwon C.K., Annamalah S.., Munusamy J., 2013. Does Marketing Mix Still Relevant? A Study on Herbal Coffee in Malaysia. International Journal of Management and Innovation (or ISO Abbreviation – Int. J. Manag. Innov.) 5, 1, 31-45.

The mentioning of the authors in the list of references in alphabetical order, from A to Z, is also recommended: thus, the text becomes way more readable, and the cited authors are more visible and easy to find and verified. It is important because, generally, there are authors with works from different years. And another important point: usually, in the list of references, the authors must be written in this way: the name (family name) and then the first name/forename, abbreviated for men, or whole for women.

Under these circumstances, the additional mention of Digital Object Identifier (DOI) becomes optional.

The observation is valid for all the articles from the bibliographic references list that are incompletely formulated.

Moreover, I suggest that the authors consult and include the following papers in the list of references:

Bondoc I., 2016. European Regulation in the Veterinary Sanitary and Food Safety Area, a Component of the European Policies on the Safety of Food Products and the Protection of Consumer Interests: A 2007 Retrospective. Part One: the Role of European Institutions in Laying Down and Passing Laws Specific to the Veterinary Sanitary and Food Safety Area. Universul Juridic, Supliment, 12-15 (Available online: http://revista.universuljuridic.ro/supliment/european-regulation-veterinary-sanitary-food-safety-area-component-european-policies-safety-food-products-protection-consumer-interests-2007-retrospective/).

Bondoc I., 2016. European Regulation in the Veterinary Sanitary and Food Safety Area, a Component of the European Policies on the Safety of Food Products and the Protection of Consumer Interests: A 2007 Retrospective. Part Two: Regulations. Universul Juridic, Supliment, 16-19 (Available online: http://revista.universuljuridic.ro/supliment/european-regulation-veterinary-sanitary-food-safety-area-component-european-policies-safety-food-products-protection-consumer-interests-2007-retrospective-2/).

Bondoc I., 2016. European Regulation in the Veterinary Sanitary and Food Safety Area, a Component of the European Policies on the Safety of Food Products and the Protection of Consumer Interests: A 2007 Retrospective. Part Three: Directives. Universul Juridic, Supliment, 20-23 (Available online: http://revista.universuljuridic.ro/supliment/european-regulation-veterinary-sanitary-food-safety-area-component-european-policies-safety-food-products-protection-consumer-interests-2007-retrospective-part/).

Bondoc I., 2016. European Regulation in the Veterinary Sanitary and Food Safety Area, a Component of the European Policies on the Safety of Food Products and the Protection of Consumer Interests: A 2007 Retrospective. Part Four: Decisions. Universul Juridic, Supliment, 24-27 (Available online: http://revista.universuljuridic.ro/supliment/european-regulation-veterinary-sanitary-food-safety-area-component-european-policies-safety-food-products-protection-consumer-interests-2007-retrospective-part-2/).

All these papers approach the matter of food safety legislation enforced within the European Union, which usually constitutes a blueprint for the law in third countries. The four documents outline the European legislative environment, starting with the year 2007, the year of the penultimate geo-political enlargement of the European Union. I want to add that all four recommended papers have been indexed in CAB International and HeinOnline, the largest and most extensive worldwide database for documents in the legal field.

The authors should pay more attention to the use of certain abbreviations to avoid confusion; basically, all abbreviations are to be used in the text-only after at least one mention made in extenso. Usually, the acronym is presented in the text after the first use of expression in extenso.

The obtained results are interpreted correctly, and their practical value is visible: however, some data are not clearly expressed in the text, situations that need to be remedied by the authors.

The graphical representation of the results is adequate; as for the grammar of the paper, most of the text is very well written, with very few parts that would require some modifications – just a few small suggestions, as follows:

Page 1, line 11 - replace ‘’perceptions of’’ with ‘’perceptions upon’’;

Page 1, line 20 – replace ‘’in Asian Pacific Region’’ with ‘’in the Asian Pacific region’’;

Page 1, line 29 – replace ‘’in Chinese’’ with ‘’in the Chinese’’;

Page 1, line 42 – replace ‘’for container’’ with ‘’for the container’’;

Page 2, line 73 – replace ‘’between conventional’’ with ‘’between the conventional’’;

Page 4, line 110 – modify ‘’(…) foods”, a series (…).’’

Page 4, line 134 – replace ‘’2’’ with ‘’two’’;

Page 5, line 152 – replace ‘’claims framework’’ with ‘’claimed framework’’;

Page 6, line 181 – replace ‘’purposes’’ with ‘’purpose’’;

Page 6, line 200 – replace ‘’to register’’ with ‘’to be registered’’;

Page 8, line 229 – ‘’to Chinese’’ with ‘’to the Chinese’’;

Page 8, line 245 – replace ‘’all functional’’ with ‘’all of the functional’’;

Page 11, line 310 - replace ‘’perceptions of’’ with ‘’perceptions upon’’;

Page 12, line 331 – replace ‘’research were’’ with ‘’research was’’;

Page 14, line 393 – replace ‘’beverages was’’ with ‘’beverages were’’;

Page 14, line 399 - replace ‘’perceptions of’’ with ‘’perceptions upon’’;

Page 14, line 400 - replace ‘’in Asian Pacific Region’’ with ‘’in the Asian Pacific region’’.

Page 14, line 400 – replace ‘’to industry’’ with ‘’to the industry’’.

The article itself, as any other article, has certain improvable aspects. By these aspects, I mean the major constituting parts of the article, as mentioned earlier, but also some elements that are related to details or writing. For example, more attention is required when presenting abbreviations, journal names, numbering of figures, or other information.

However, the overall paper presents a high degree of originality, making it attractive to academic staff, to the researchers in the field, and even to the broad public.

Minor corrections and clarifications notwithstanding, the authors’ work and obtained results are highly commendable. They bring significant added value to the work and may constitute a launching pad for further useful studies.

Provided that the authors verify the paper and perform the required corrections, the article may be published in the Beverages.

Best Regards,

Reviewer

Author Response

We would like to take this opportunity to thank the Editor and Reviewers for their useful suggestions on how to improve our manuscript. We are confident that this version is much improved as a result of their feedback and comments and hope you agree so too. 

N.B.

The line numbers are based on the revised tracked version (with “all makeup”).

Comment 1:

The bibliography is relevant, but presents some minor lacks when it comes to citations or mentions. To clarify some aspects, I would suggest that the authors write the bibliography evenly: for example, journal papers require either the complete journal name, or the JCR abbreviation (in the case of ISI indexed or rated journals), or the ISO abbreviation (for BDI indexed journals); moreover, for journals, I suggest that the volume, number, and pages (as the case requires) be mentioned.

The observation is valid for all the articles from the bibliographic references list that are incompletely formulated.

Response:

The references have been checked carefully and the format now follows the requirements of the journal Beverages.

Comment 2:

Moreover, I suggest that the authors consult and include the following papers in the list of references:

All these papers approach the matter of food safety legislation enforced within the European Union, which usually constitutes a blueprint for the law in third countries. The four documents outline the European legislative environment, starting with the year 2007, the year of the penultimate geo-political enlargement of the European Union. I want to add that all four recommended papers have been indexed in CAB International and HeinOnline, the largest and most extensive worldwide database for documents in the legal field.

Response:

The suggested references have been cited in the manuscript. See Lines 429-440 and below:

A challenge with functional beverages is that owing to differences in legislation between   countries, a single product may be categorized and consequently regulated differently in two countries. Therefore, professional interpretation of regulations is critical as is effective communication between regulators and product manufacturers in order to ensure that regulations are adhered to. For example, the detailed interpretation of European policies on the safety of food products and the protection of consumer interests by Bondoc I. 2016 has played a positive role in the enforcement of food safety legislation within the European Union, especially in the Romanian region [57-60]. Similar research within the Asia Pacific Region is recommended.

Bondoc I., 2016. European Regulation in the Veterinary Sanitary and Food Safety Area, a Component of the European Policies on the Safety of Food Products and the Protection of Consumer Interests: A 2007 Retrospective. Part One: the Role of European Institutions in Laying Down and Passing Laws Specific to the Veterinary Sanitary and Food Safety Area. Universul Juridic, Supliment, 12-15. Available online: http://revista.universuljuridic.ro/supliment/european-regulation-veterinary-sanitary-food-safety-area-component-european-policies-safety-food-products-protection-consumer-interests-2007-retrospective/ (accessed 2 Feb 2020). Bondoc I., 2016. European Regulation in the Veterinary Sanitary and Food Safety Area, a Component of the European Policies on the Safety of Food Products and the Protection of Consumer Interests: A 2007 Retrospective. Part Two: Regulations. Universul Juridic, Supliment, 16-19. Available online: http://revista.universuljuridic.ro/supliment/european-regulation-veterinary-sanitary-food-safety-area-component-european-policies-safety-food-products-protection-consumer-interests-2007-retrospective-2/ (accessed 2 Feb 2020). Bondoc I., 2016. European Regulation in the Veterinary Sanitary and Food Safety Area, a Component of the European Policies on the Safety of Food Products and the Protection of Consumer Interests: A 2007 Retrospective. Part Three: Directives. Universul Juridic, Supliment, 20-23. Available online: http://revista.universuljuridic.ro/supliment/european-regulation-veterinary-sanitary-food-safety-area-component-european-policies-safety-food-products-protection-consumer-interests-2007-retrospective-part/ (accessed 2 Feb 2020). Bondoc I., 2016. European Regulation in the Veterinary Sanitary and Food Safety Area, a Component of the European Policies on the Safety of Food Products and the Protection of Consumer Interests: A 2007 Retrospective. Part Four: Decisions. Universul Juridic, Supliment, 24-27. Available online: http://revista.universuljuridic.ro/supliment/european-regulation-veterinary-sanitary-food-safety-area-component-european-policies-safety-food-products-protection-consumer-interests-2007-retrospective-part-2/ (accessed 2 Feb 2020).

Comment 3:

The authors should pay more attention to the use of certain abbreviations to avoid confusion; basically, all abbreviations are to be used in the text-only after at least one mention made in extenso. Usually, the acronym is presented in the text after the first use of expression in extenso.

Response:

The abbreviations have been checked and amended carefully.

Comment 4:

The graphical representation of the results is adequate; as for the grammar of the paper, most of the text is very well written, with very few parts that would require some modifications – just a few small suggestions, as follows…

Response:

The grammar corrections have been made based on reviewers’ comments.